genetics/plant science

cytoplasmic male sterility, plant reproduction, nuclear–mitochondrial interaction, gene duplication, neofunctionalization

**Author for correspondence:**
Tomohiko Kubo
e-mail: tomohiko@abs.agr.hokudai.ac.jp

†Present address: Gifu Prefectural Research Institute for Agricultural Technology in Hilly and Mountainous Areas, Nakatsugawa, Gifu 508-0203, Japan.

# How did a duplicated gene copy evolve into a *restorer-of-fertility* gene in a plant? The case of *Oma1*

Takumi Arakawa[1,†], Hajime Sugaya[1],
Takaya Katsuyama[1], Yujiro Honma[1,2], Katsunori Matsui[1],
Hiroaki Matsuhira[3], Yosuke Kuroda[3], Kazuyoshi Kitazaki[1]
and Tomohiko Kubo[1]

[1]Research Faculty of Agriculture, Hokkaido University, Sapporo, Hokkaido 060-8589, Japan
[2]Department of Biotechnology and Environmental Chemistry, Kitami Institute of Technology, Kitami, Hokkaido 090-8507, Japan
[3]Hokkaido Agricultural Research Center, National Agriculture and Food Research Organization, Memuro, Hokkaido 082-0081, Japan

TK, 0000-0001-6045-0167

*Restorer-of-fertility* (*Rf*) is a suppressor of cytoplasmic male sterility (CMS), a mitochondrion-encoded trait that has been reported in many plant species. The occurrence of CMS is considered to be independent in each lineage; hence, the question of how *Rf* evolved was raised. Sugar beet *Rf* resembles *Oma1*, a gene for quality control of the mitochondrial inner membrane. *Oma1* homologues comprise a small gene family in the sugar beet genome, unlike *Arabidopsis* and other eukaryotes. The sugar beet sequence that best matched *Arabidopsis atOma1* was named *bvOma1*; sugar beet *Rf* (*RF1-Oma1*) was another member. During anther development, *atOma1* mRNA was detected from the tetrad to the microspore stages, whereas *bvOma1* mRNA was detected at the microspore stage and *RF1-Oma1* mRNA was detected during the meiosis and tetrad stages. A transgenic study revealed that, whereas *RF1-Oma1* can bind to a CMS-specific protein and alter the higher-order structure of the CMS-specific protein complex, neither *bvOma1* nor *atOma1* show such activity. We favour the hypothesis that an ancestral *Oma1* gene duplicated to form a small gene family, and that one of the copies evolved and acquired a novel expression pattern and protein function as an *Rf*, i.e. *RF1-Oma1* evolved via neofunctionalization.

# 1. Introduction

In plants, hermaphrodites are converted into females by male sterility encoded by mitochondria, a phenomenon known as cytoplasmic male sterility (CMS) [1,2]. Mitochondrial genes responsible for CMS (*S-orf*) are composed of partial duplicates of ordinary genes (such as those coding for ATP synthase subunits) and/or origin-unknown sequences [3,4]; their primary structure varies among plant species, suggesting that each incidence of CMS has an independent origin. From the viewpoint of evolutionary genetics, the maternal inheritance of mitochondria favours the evolution of CMS because the resources for pollen production can be saved and used to increase female fitness. On the other hand, CMS causes a genetic conflict between the mitochondria and the nuclear genome (biparental inheritance), and the decrease in pollen transmission due to CMS creates pressure for the evolution of a counteracting system in the nuclear genome [2,5].

Genetic analysis of the CMS suppression system revealed a nuclear gene termed *restorer-of-fertility* (*Rf*) [6–8]. A dominant *Rf* allele suppresses *S-orf*, thereby restoring pollen fertility [9]. Differences in CMS have been defined by differences among the cognate *Rf* genes [10], implying genetic diversity among *Rf*. The proposed evolutionary mechanism for *Rf* has implicated a molecular arms race, analogous to the coevolution of pathogens and resistance genes [11–13]; however, the initial steps of *Rf* evolution are obscure at the molecular level.

Sugar beet (*Beta vulgaris* L.) CMS involves *preSatp6* as the *S-orf* and *Rf1* as the *Rf* [14,15]. *preSatp6*, named after a unique presequence in CMS mitochondrial *atp6*, is composed of a 387 amino acid sequence of unknown origin. Translation products from *preSatp6* were detected in all examined organs to form a 250 kDa protein complex in the mitochondrial membrane of CMS plants [16], although this complex's function is unknown. When the plant has a dominant *Rf1* allele, preSATP6 protein in the anther is detected in a novel 200 kDa complex concomitantly with a decrease in the amount of the 250 kDa complex, whereas the total amount of preSATP6 protein is almost unchanged [16,17]. This result suggests the anther-specific alteration of a higher-order structure of the preSATP6 protein.

The *Rf1* locus consists of a gene cluster that shows copy number variation (CNV) among breeding lines [15,18,19]. Genes in the cluster have homology to *Oma1*, a gene encoding an ATP-independent protease that participates in quality control of the mitochondrial inner membrane (yeast) and mitochondrial dynamics (mammals) [20–22]. In Arabidopsis, *Oma1* is a single copy gene (hereafter *atOma1*); loss of *atOma1* causes some disorder in oxidative phosphorylation (OXPHOS) [23].

A puzzling observation was seen in the clustered *Oma1*-like genes in the *Rf1* locus. The zinc-binding motifs in their M48 peptidase domains, whether dominant or recessive, were identical to that of a mutagenized gene that lost proteolytic activity in yeast (i.e. H**Q**VGH instead of the consensus H**E**xxH) [15]. Therefore, we hypothesized that there might be another *Oma1* homologue that functions as the authentic *Oma1* in sugar beet. If so, identifying this sequence will facilitate determining how *Oma1*-like genes at the *Rf1* locus (hereafter *RF1-Oma1*) evolved from *Oma1*. We found that another *Oma1*-like gene, with the consensus HExxH motif, is preserved in the sugar beet genome; however, its translation product is incapable of binding preSATP6 protein or generating the 200 kDa protein complex. Moreover, its expression pattern is different from that of *RF1-Oma1*. We propose that in sugar beet, *Rf1* evolved via neofunctionalization.

# 2. Material and methods

## 2.1. Bioinformatic analysis

Nucleotide sequences were retrieved from the NCBI website (https://www.ncbi.nlm.nih.gov/assembly/). Database searching was conducted at the NCBI website (https://blast.ncbi.nlm.nih.gov/Blast.cgi). Alignment of nucleotide and amino acid sequences was done using ClustalW (http://clustalw.ddbj. nig.ac.jp/index.php?lang=ja) and MEGA (https://www.megasoftware.net) algorithms [24]. The alignment was visually inspected and modified manually. The microarray-based expression pattern of *atOma1* was retrieved from a website (http://bar.utoronto.ca/efp2/Arabidopsis/Arabidopsis_eFPBrowser2.html) [25] using At5g51740.1 as the query.

## 2.2. Plant materials

The beet lines used in this study are summarized in table 1. Sugar beet (*Beta vulgaris* ssp. *vulgaris*) lines TA-33BB-O, TA-33BB-CMS, TK-81 mm-O, NK-198, NK-219 mm-O, NK-219 mm-CMS and NK-305 were

**Table 1.** Beet lines used or mentioned in this study.

| name of line | cytoplasm[a] | nuclear genotype | origin[b] |
|---|---|---|---|
| TA-33BB-O | N | rf1rf1 | NARO |
| TA-33BB-CMS | S | rf1rf1 | NARO |
| TK-81 mm-O | N | rf1rf1 | NARO |
| NK-198 | S | Rf1Rf1 | NARO |
| NK-219 mm-O | N | rf1rf1 | NARO |
| NK-219 mm-CMS | S | rf1rf1 | NARO |
| NK-305 | S | Rf1Rf1 | NARO |
| PI 507186 | unknown | unknown | USDA |
| KWS2320 | unknown | rf1rf1 | KWS |
| EL10 | unknown | unknown | MSU |

[a]N, male fertile; S, male sterile.
[b]NARO, National Agriculture and Food Research Organization; USDA, US Department of Agriculture; KWS, KWS Saat; MSU, Michigan State University.

developed at the Hokkaido Agricultural Research Center, National Agriculture and Food Research Organization, Japan. TA-33BB-O and TA-33BB-CMS, and NK-219 mm-O and NK-219 mm-CMS, respectively, have the same nuclear genotype but differ in their cytoplasms: suffixes '-O' and '-CMS' mean male fertile and male sterile cytoplasms, respectively. A wild beet (*B. vulgaris* ssp. *maritima*) accession PI 507186 was obtained from the United States Department of Agriculture. Seeds of Arabidopsis (*Arabidopsis thaliana*) ecotype Col-0 were a gift from Prof. Dr Satoshi Naito (Hokkaido University). All plants were grown in a greenhouse. Crosses were made as described in [26]. TA-33BB-CMS(NK-198 *Rf1*) is a BC6F1 derived from a cross of TA-33BB-CMS x NK-198.

## 2.3. Genotyping

Total cellular DNA was isolated from fresh green leaves by the CTAB method according to the procedure of [27]. PCR was performed with primers #1 and #2 (electronic supplementary material, table S1). GoTaq Green Master Mix (Promega, Madison, WI, USA) was used for genotyping. PCR cycles were: 1× 98°C 1 min; 35× (95°C 30 s, 55°C 30 s and 72°C 1 min 30 s) and 1× 72°C 3 min. PCR products were electrophoresed in 2% agarose gels.

## 2.4. Quantitative reverse transcription–PCR

Excised organs were frozen in liquid nitrogen and then powdered using a Multi-Beads Shocker (Yasui Kikai, Osaka, Japan). Total cellular RNA was isolated using an RNeasy Plant Mini Kit (Qiagen, Valencia, CA, USA). Residual genomic DNA was digested with RNase-free DNase (Promega). Procedures for cDNA synthesis and quantification were followed as described in [17]. The primers for each gene were: #3 and #4 for *bvOma1*, #5 and #6 for *RF1-Oma1*, #7 and #8 for *Actin* and #9 and #10 for EF1$\alpha$ (see electronic supplementary material, table S1). The specificity of the *bvOma1* and *RF1-Oma1* primers was verified by PCR with plasmids carrying the target sequences (electronic supplementary material, figure S1).

## 2.5. *In situ* hybridization

Procedures for *in situ* hybridization (ISH) followed the protocols outlined in the Cold Spring Harbor Arabidopsis Genetics Course (https://www.arabidopsis.org/cshl-course/5-in_situ.html) and [28]. Hybridization probes were generated by *in vitro* transcription of a pBluescript SK-cloned DNA fragment using a DIG RNA Labeling Kit (Roche Diagnostics, Mannheim, Germany). DNA fragments for probes were generated by PCR using PrimeSTAR Max (Takara Bio, Kusatsu, Japan) as the DNA polymerase and total cellular DNA of TA-33BB-O or Col-0 as templates. The primers for each gene were: #11 and #12 for *atOma1*, #13 and #14 for *bvOma1* and #15 and #16 for *RF1-Oma1* (see electronic

supplementary material, table S1). Hybridized sections were observed using a BX50 light microscope equipped with a DP21 CCD camera (Olympus, Tokyo, Japan).

## 2.6. Transgene construction

Complementary DNAs were synthesized from total cellular RNAs isolated from fresh green leaves of TA-33BB-O or Col-0. The coding regions of *bvOma1* and *atOma1* were amplified by PCR using the primers #17 and #18 for *bvOma1* and #19 and #20 for *atOma1* (see electronic supplementary material, table S1). The resultant PCR fragments were cloned into pDONR/zeo via the Gateway system (Thermo Fisher Scientific, Waltham, MA, USA). A FLAG tag was added by using a PrimeSTAR Mutagenesis Basal Kit (Takara Bio) with the primers #21 and #22 for *bvOma1::FLAG* and #23 and #24 for *atOma1::FLAG* (see electronic supplementary material, table S1). The cloned fragments were introduced into a binary vector pMDCΩ [16] via the Gateway system. *RF1-Oma1::FLAG* is the same as *bvORF20::flag* that was reported in [16].

## 2.7. Transgenic suspension cells

The sugar beet line NK-219 mm-CMS was chosen for transformation because it has been used previously in transgene experiments [29]. Transgenic suspension cells were generated by Agrobacterum-mediated transformation [16].

## 2.8. Isolation of crude mitochondria

About 100–200 mg of suspension cells were ground in an extraction buffer (50 mM Tris–HCl (pH 8.0), 0.5 M mannitol, 1 mM EDTA-Na$_2$, 0.1% (w/v) BSA, 1.0% sodium-L-ascorbate and 0.5% (w/v) Polyclar AT) with a plastic pestle, and centrifuged (5500$g$ at 4°C for 10 min). The supernatant was recentrifuged (6500$g$ at 4°C for 10 min), and the resultant supernatant was transferred to a new tube and centrifuged again (11 000$g$ at 4°C for 15 min). The pellet was washed in a wash buffer (50 mM Tris–HCl (pH 7.4), 0.5 M mannitol and 1 mM EDTA-Na$_2$). After centrifugation (11 000$g$ at 4°C for 15 min), the pellet of crude mitochondria was suspended in the wash buffer.

## 2.9. Blue-native polyacrylamide gel electrophoresis

Suspended mitochondria were added to an equal volume of 2× NativePAGE Sample Buffer (Thermo Fisher Scientific) containing 2% (w/v) digitonin, and left on ice for 30 min. After centrifugation (11 500$g$ at 4°C for 15 min), the supernatant was applied to a NativePAGE Novex 4–16% Bis-Tris Gel (Thermo Fisher Scientific) for electrophoresis following the manufacturer's instructions.

## 2.10. Co-immunoprecipitation

About 40 μg of crude mitochondria was suspended in 1× PBS containing Protein Inhibitor Cocktail for Plant Cell and Tissue Extracts (Sigma, St Louis, MO, USA) and 2.5% (w/v) digitonin, and left at 4°C for 30 min. After centrifugation (11 000$g$ at 4°C for 15 min), the supernatant was collected. The digitonin concentration was adjusted to 0.1% (w/v) by adding 1× PBS, and the volume of the sample was estimated. Anti-DDDDK Tag Magnetic Beads were first equilibrated in 1× PBS containing 0.1% (w/v) digitonin, then added to the sample in the ratio of approximately 4 μl bead slurry/sample as outlined in the instruction manual for the DDDDK-tagged Protein Magnetic Purification Kit (MBL, Nagoya Japan). Samples were end-over-end mixed overnight at 4°C. Beads were collected using a Magnetic Rack and washed twice with 1× PBS containing 0.1% (w/v) digitonin. Immunoprecipitates were eluted by boiling the beads in the SDS sample buffer (see §2.11).

## 2.11. SDS–polyacrylamide gel electrophoresis

Samples were suspended in SDS sample buffer (50 mM Tris–HCl (pH 6.8), 2% (w/v) SDS, 10% glycerol, 0.005% (w/v) bromophenol blue and 1% β-mercaptoethanol), and boiled for 5 min. Electrophoresis was conducted using 12% SDS–polyacrylamide gels after the method of Schägger & von Jagow [30].

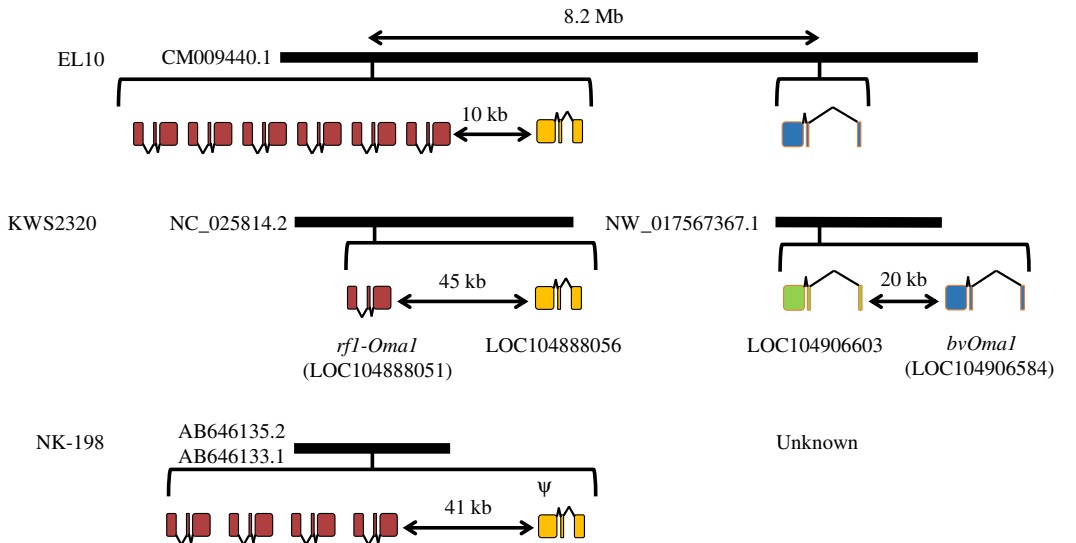

**Figure 1.** Variation of *Oma1* homologues in three sugar beet lines, EL10, KWS2320 and NK-198. Bold horizontal lines represent chromosomal segments and are identified by NCBI reference sequence numbers or GenBank/EMBL/DDBJ accession numbers. Brackets indicate gene loci. Boxes and wedges show exons and introns, respectively. Gene direction is coordinated with that of EL10: *bvOma1*, *LOC104906603* and *LOC104888056* are transcribed from left to right and *RF1-Oma1* is from right to left. Colours of exons indicate cognate genes: blue, *bvOma1*; green, *LOC104906603*; red, *RF1-Oma1* (or *rf1-Oma1*); and yellow, *LOC104888056*. *LOC104888056* of NK-198 is apparently a pseudogene due to the presence of a frame shift mutation (indicated by ψ). Length of intervals is shown by double-headed arrows. Note that line lengths are not proportional. The nucleotide sequence of *bvOma1* in NK-198 is unknown (but see electronic supplementary material, figure S3).

## 2.12. Protein gel blot analysis

Separated proteins were blotted onto Hybond-P PVDF membranes (GE Healthcare, Little Chalfont, UK) with a Mini TransBlot Cell (Bio-Rad Laboratories, Hercules, CA, USA). Can Get Signal system (Toyobo, Osaka, Japan) was used for antibody/antiserum reaction. Primary antibodies/antisera included a mouse monoclonal anti-DDDDK (anti-FLAG) (MBL), and rabbit anti-preSATP6 and anti-COXI [14]. Secondary antibodies were HRP-conjugated goat anti-mouse IgG and HRP-conjugated goat anti-rabbit IgG (Jackson Immunoresearch, West Grove, PA, USA). ECL Prime (GE Healthcare) was used as the substrate for chemiluminescence. Signal bands were detected on X-ray film (GE Healthcare).

# 3. Results and discussion

## 3.1. *Oma1* homologues in the sugar beet genome

Two high-quality sugar beet genome sequences were publicly available, one from the sugar beet line KWS2320 and the other from EL10 [31,32]. We searched for *Oma1* homologues in these genomes using the tBLASTN algorithm with the amino acid sequence of *atOma1* (At5g51740.1) as the query sequence. We identified four and eight sequences in KWS2320 and EL10, respectively (electronic supplementary material, table S2). All 12 of these sequences were predicted to be interrupted by two introns as is the case for *atOma1* (figure 1).

The best matched gene to *atOma1* in KWS2320 was *LOC104906584* on a scaffold (NCBI reference sequence NW_017567367.1). This gene was tentatively named *bvOma* because it is the only copy having the consensus zinc-binding motif of an M48 peptidase domain (HExxH) (figure 2). This scaffold contained another *Oma1* homologue, *LOC104906603*, that is 20 kb distant from *bvOma1*; however, when compared with *bvOma1*, *LOC104906603* had 153 and 57 bp deletions in its first exon and a 6 bp deletion and a 21 bp insertion in the second exon (electronic supplementary material, figure S2). Shared amino acid residues between *atOma1* and *bvOma1* were lost by these insertions/ deletions (figure 2). The other two *Oma1*-homologous genes (*LOC104888051* and *LOC104888056*) were on another scaffold (NC_025814.2) and had less homology to *atOma1*. We noticed that *LOC104888051* was identical to *rf1-Oma1* from a recessive *rf1* allele (we use *rf1-Oma1* when it is known to be

```
atOma1          MSWYRRTKLV FDSLRRNINP KILPRSHVTS RINNPIGSSN PSAKFSS--I 48
bvOma1          MAWYRRSRFV YNAYKSLNSK LLLPKSPVQS PI--PRFNSN SSSLFYNQFK 48
LOC104906603    MAWYRRSRFV YNAYKSLNSK LLLPKSPVPS PV--PRINSN SSSLFYNQFK 48
rf1-Oma1        MAWYRNSRFV YNALKLNLRS ----KTFGTI PT--PRVHSN SSSLFYN--Q 42
LOC104888056    MAFHRNSRFV YNALKPSFNS ----KLLTKT SS-----HSN SYSLFYTQFK 41

atOma1          SSREVGLRSW TSLGRNTNRI AYNP---FLS Q--PKRYYYV DRYQVRHFKP 93
bvOma1          SSIISGSPSI SSKFGYLNGV KQNQ-SSLFS C-VTRRNYHV DRNQIYHFKP 96
LOC104906603    SSIISGSPSI SSKFGYLNGV KQNQ-SSLFS C-VTRRNYHV DTNQIYLF-- 94
rf1-Oma1        ST-KCSGLFG SAKSGYFNGF KHHQEISSFS G-FARRNYHG VKTEVSVEF- 89
LOC104888056    YSRLHGSPSI SSKCGYFNGF KHTQ-NRIIS GVATIRNSLE VKKTQKFYG- 89

atOma1          RGPGRWFQNP RTVFTVVLVG SVGLITLIVG NTETIPYTKR THFILLSKPM 143
bvOma1          RGFKSWFENP RHIFIAVVIG SGVVITVYFG NSEVVPYTKR KHLVLLSRTL 146
LOC104906603    ---------- ---------- ---------- ---------- --------L 95
rf1-Oma1        ------RVEK LLLGIALIIS HSGMIAFFYL HPVVVPYTGR KHYVILSTTH 133
LOC104888056    -------KQL KKTNSWWVDG IFFVMTIYYS ISEVVPFTER KHLVIPLTSL 133

atOma1          EKLLGETQFE QIKKTYQGKI LPATHPESIR VRLIAKEVID ALQRGLSNER 193
bvOma1          ERRIGDSQFE KMKEEFKGKI LPAIHPDSVR VRLISKDIIE SLERGISHER 196
LOC104906603    ERRYGEFRFE KRKEDFKGKI LPAIHPDSVR VRLISKDIIE SLGRGISHER 145
rf1-Oma1        ENENGE--FE KRK------I QPATHPDTER VRSIFQHILE SLEREINHHE 175
LOC104888056    EIKIGES--- MKKKLYDGKT LHARHPASVR ARVVFEHIIV SLDHKLIHE- 179

atOma1          VWSDLGYAST ESSLGGGSDK GVKEMEMAMS GEDTMTDMKW SKEDQVLDDQ 243
bvOma1          AWSSPGYA-T ESV-SHHEID GHETMKALTE GMDEKVPGDW HKEEEVLDDK 244
LOC104906603    AWST------ ---------- -----KALTE GMDEKVPRDW HKEEEVLDDK 174
rf1-Oma1        LELELELE-- ---------- ---------- RDETFKEKTI WKEETDHD-- 201
LOC104888056    ---------- ---------- ---------- ---------- ---------- 179

atOma1          WIQKSRKKDS K--AHAATSH LEGISWEVLV VNEPIVNAFC LPAGKIVVFT 291
bvOma1          WVKDSRKKGE KHGAKTTTNH LEGLNWEVLV VNEPVVNAFC LPGGKIVVFT 294
LOC104906603    WVKDSRKKGE KHGAKTTTNH LEGLNWEVLV VNEPFVNASY FPGGKIVVFT 224
rf1-Oma1        --KDSRKKHS G--AKITTNH -EGMNWEIFV VDKPWVESSC IFGGKIVVYT 246
LOC104888056    --------GN G--SKTTTKH -----LEVFV VDEPRVFSFC FPGGMIAVST 214

atOma1          GLLNHFKSDA EVATVIC HEV GH AVARHVAE GITKNLW--- -FAILQLVLY 337
bvOma1          GLLKHFKSDA ELATIIC HEV GH AVARHSAE QITKNMW--- -FAILQLILY 340
LOC104906603    GLLKHLKSDA ELATIIC HEV GH AVARHSAE RITWIMS--- -FASLQLIL- 269
rf1-Oma1        GLLNHCISDA ELATIIA HQV GH AVARHEAE HWTTLLWSIL LVIYMTIFQY 296
LOC104888056    GLLNYFHSDS ELAAIIC TQV AD AVARPFAE FFPKYML--- AMFVISIIN- 260

atOma1          QFVMPDLVNT ----MSA--- LFLRIIP---- ---------- ---------- 356
bvOma1          QFIAPDFANA ----MSN--- LLLRIIP---- ---------- ---------- 359
LOC104906603    -LIALDFAYA RYLQISSYVG VLLIILS---- ---------- ---------- 294
rf1-Oma1        LFTAPEFANA IS-------K LLSRHPLLQK VWKIIQARFH QLLPRTTLHL 339
LOC104888056    ---------- ---------- --PSAR---- ---------- ---------- 264

atOma1          ---------- ---FSRKMEI EADYIGLLLL ASAGYDPRVA PTVYEKLGKL 393
bvOma1          ---------- ---FSRKMEI EADYIGLLLM ASAGYDPRIA PQVYEKLGKI 396
LOC104906603    ---------- ---FDRKGEI EADYIGLLLM ASAGYDPRIA PQVYEKLGKI 331
rf1-Oma1        GFLGLSSLVF ILYFGRK-EI EADHIGVLLM ASAGYDPRVA PQVYDKLAKP 388
LOC104888056    --------IV KIIQARACKL RP-FTAGLIK SGLNFTG-LL LLCFAPLDCY 304

atOma1          GGD--ALGDY LSTHPSGKKR SKLLAQANVM EEALMIYREV QAGRTGVEGF 441
bvOma1          SGESSSLTEY LSTHPSGKKR AQLLARAHIM QEAVDMYREI VAGRA-IEGF 445
LOC104906603    SGESSSLKEY LSTHPSGKKR AQLLARAHIM KEAVDIYREI VAGHA-IEGF 380
rf1-Oma1        LGD----WNC LATHPFARMR AKLLARADVM KEADKIYNEV VAGRA-IQGL 433
LOC104888056    FRWRKMEADY IGLQLMSSAG ----YDPRVA PQAYQKLRRQ TIA------- 343

atOma1          L 442
bvOma1          L 446
LOC104906603    L 381
rf1-Oma1        Q 434
LOC104888056    - 343
```

**Figure 2.** Alignment of amino acid sequences deduced from *atOma1*, *bvOma1* (KWS2320), *LOC104906603* (KWS2320), *rf1-Oma1* (KWS2320) and *LOC104888056* (KWS2320). Residues are numbered from the first methionine. Dashes are incorporated for maximum matching. Zinc-binding motifs are enclosed in a box, and positions of introns are shown by triangles and dashed lines.

recessive) [15]. *LOC104888056* had the least homology to *atOma1*. The *rf1-Oma1* and *LOC104888056* were 45 kb apart. Both the NW_017567367.1 and NC_025814.2 scaffolds were assigned to chromosome 3, but their physical locations on the chromosome are unknown.

In EL10, all *Oma1* homologues were located on a contig from chromosome 3 (GenBank accession number CM009440.1) (figure 1). Whereas *bvOma1* was identified, *LOC104906603* was not present in the EL10 genome, suggesting polymorphism in the presence/absence of *LOC104906603* among sugar beet lines. This supposition was confirmed by the analysis of our beet collections (electronic supplementary material, figure S3). We found the *Rf1* locus of EL10 consisted of six copies of *RF1-Oma1*. *LOC104888056* of EL10 was identified in a region 10 kb apart from the *Rf1* locus. Copies of *RF1-Oma1* were on the opposite strand of other *Oma1* homologues.

A 383 kb genomic region containing the *Rf1* locus of sugar beet line NK-198 was previously described [15]. We found the NK-198 counterpart of *LOC104888056*, but it had an 8 b insertion in the first exon that led to a frame shift mutation (electronic supplementary material, figure S4).

Our previous reports indicated that no sugar beet line examined so far has lost *rf1-Oma1* [18,26], and in this study, we found that *bvOma1* and *RF1-Oma1* are ubiquitous among all examined lines. Additional *Oma1*-like duplicates occur, but they are not always conserved among sugar beet lines; hence, we focused our analysis on *bvOma1* and *RF1-Oma1*.

## 3.2. Differences in the spatial and temporal expression patterns between *Oma1* homologues

The expression patterns of *bvOma1*, *RF1-Oma1* and *atOma1* were compared. We considered *atOma1* to be an outgroup *Oma1* for the beet *Oma1*-like genes, because *atOma1* is the only *Oma1* homologue in Arabidopsis, and it can complement the deficiency of yeast *Oma1* [23]. Web-retrieved microarray-based expression data of *atOma1* indicated the detection of *atOma1* mRNA from almost all organ/tissues, whereas increased expression was seen in young stamens, mature pollen, immature seeds and dry seeds (expression levels 2–9 times that of root) (electronic supplementary material, figure S5).

We analysed *bvOma1* expression in the sugar beet line TA-33BB-O, in which no *LOC104906603* was seen (electronic supplementary material, figure S3). Quantitative reverse transcription–PCR (qRT–PCR) detected *bvOma1* mRNA from roots, green leaves, peduncles without flowers, immature anthers and other floral organs (electronic supplementary material, table S3). Highest expression was measured in immature anthers (three times that of root). We also quantified the mRNA of *rf1-Oma1* in TA-33BB-O, which is identical to the KWS2320 *rf1-Oma1*. Although *rf1-Oma1* is unable to restore pollen fertility, its open reading frame (ORF) is uninterrupted and is highly conserved among sugar beet lines [18]. The transcripts were relatively abundant in immature anthers compared to other organs (473–487 times that of root).

We focused on the expression patterns of *atOma1*, *bvOma1* and *RF1-Oma1* in anthers. We used the Arabidopsis accession Col-0 for ISH of *atOma1*. No signal was seen at the meiosis stage (figure 3*a*). Sections with tetrads gave signals within the tapetal tissues, the inner-most cell layer of anther locules and tetrads (figure 3*a*). Sections with microspores produced signals within both tapetal tissues and microspores (figure 3*a*). A sense-probe yielded signals within microspores.

We conducted ISH using sugar beet line TA-33BB-CMS(NK-198 *Rf1*), which has S mitochondria but is male fertile due to a dominant *Rf1* introduced from NK-198 by recurrent backcrossing, from which we expected a stronger signal of *RF1-Oma1* than that obtained in TA-33BB-O. Expression signals of *bvOma1* were obtained from tapetal tissues and microspores in sections of the microspore stage but hardly seen in sections at the meiotic and tetrad stages (figure 3*b*). On the other hand, *RF1-Oma1* mRNA was detected from tapetal tissues in sections at both the meiotic and tetrad stages (figure 3*c*). Faint signals were also seen in microspore mother cells, whereas the signals in tetrads were uneven, and we therefore think their significance was questionable (figure 3*c*). No signal was observed in sections at the microspore stage.

## 3.3. Examination of molecular interactions with *preSatp6*

*RF1-Oma1*, but not *rf1-Oma1*, can give rise to a 200 kDa protein complex containing preSATP6 when expressed in suspension cells of CMS sugar beet [16,17,19]. We examined whether *atOma1* and *bvOma1* had the same activity. FLAG tag-fused *atOma1* and *bvOma1* cDNAs were constructed using a virus-derived promoter for constitutive expression. The two transgenes were introduced into suspension cells of CMS sugar beet via Agrobacterium (electronic supplementary material, figure S6). Results of immunodetection of preSATP6 from protein samples resolved by Blue-native polyacrylamide gel electrophoresis (PAGE) are shown in figure 4*a*. No alteration was seen on the blot of the transgenic suspension cells expressing *atOma1* or *bvOma1*, whereas *RF1-Oma1* from NK-198 produced an immunodetected 200 kDa band (figure 4*a*).

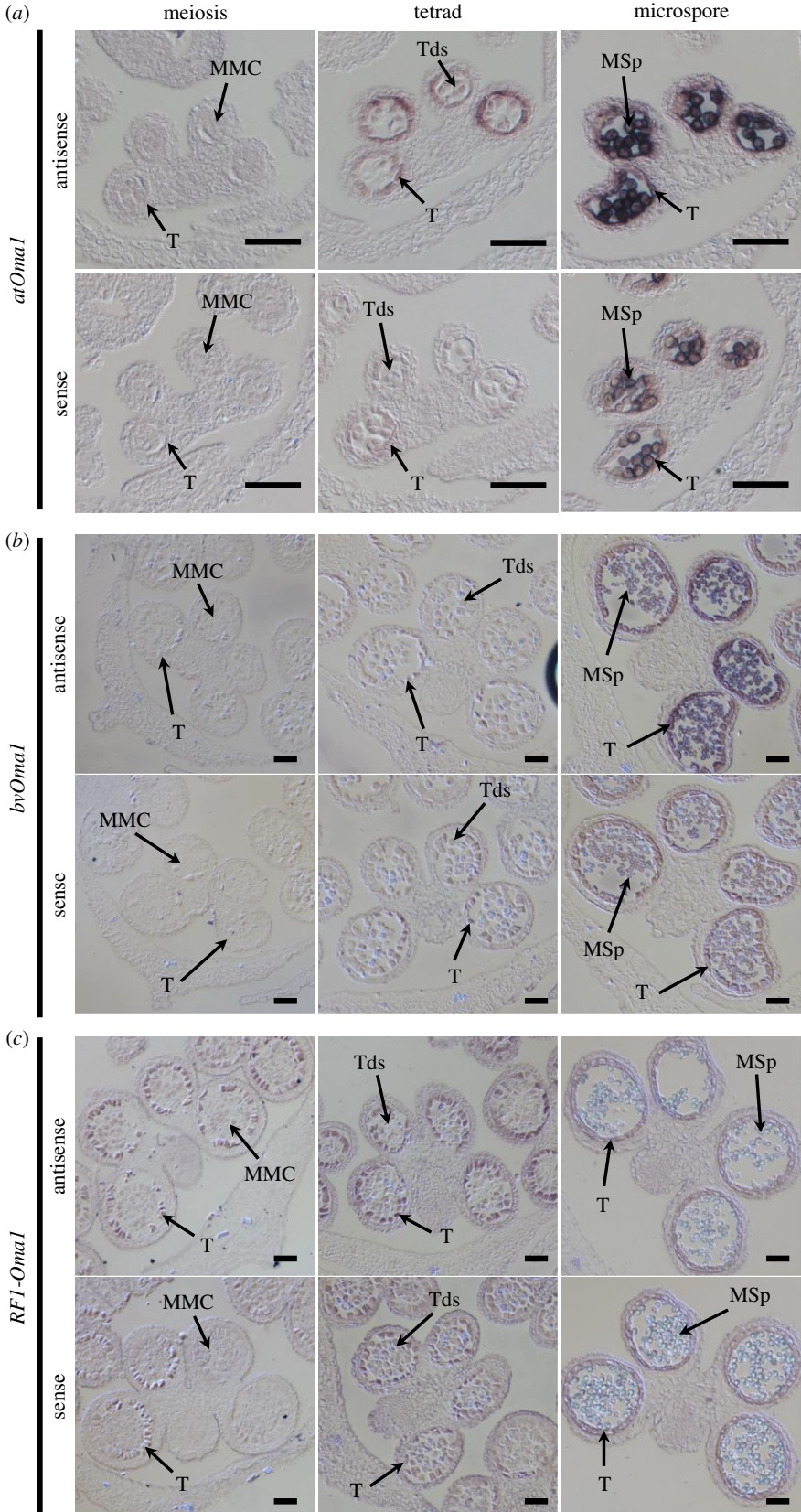

**Figure 3.** Detection of *atOma1*, *bvOma1* and *RF1-Oma1* mRNAs in anther tissues by ISH. Images of antisense and sense probes are shown. MMC, microspore mother cell; T, tapetal cell; Tds, tetrads; MSp, microspore. Scale bars are 50 μm. (*a*) *atOma1* expression in Arabidopsis Col-0 anthers. Meiosis, tetrad and microspore stages correspond to developmental stages 6, 7 and 9 of [33]. (*b,c*) *bvOma1* and *RF1-Oma1* expression in sugar beet anthers, respectively. Meiosis, tetrad and microspore stages correspond to the developmental stages of Meiosis, Tetrad and Microspore (Sb-1) described by Arakawa *et al.* [17].

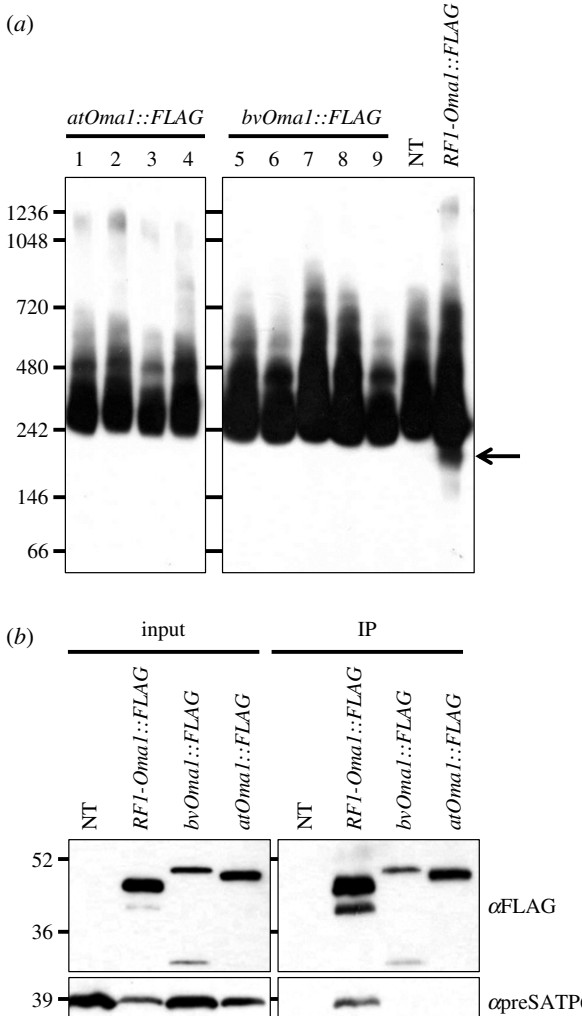

**Figure 4.** Immunoblot analysis of proteins from transgenic suspension cells of sugar beet. Size markers are shown on the left (kDa). (a) Total protein extracts of a crude mitochondrial preparation isolated from transgenic suspension cells were subjected to 4–16% Blue-native PAGE, followed by blotting of the separated protein complexes onto a membrane. The membrane was probed with anti-preSATP6. Lanes 1–4 and 5–9 are cell lines transformed with *atOma1::FLAG* and *bvOma1::FLAG*, respectively, with expression driven by the cauliflower mosaic virus (CaMV) 35S promoter (see electronic supplementary material, figure S6). NT denotes non-transgenic suspension cells. *RF1-Oma1::FLAG* is a cell line whose *RF1-Oma1* coding region in the transgene is derived from the dominant *Rf1* allele (*bvORF20* in [16]). The 200 kDa signal band that is the hallmark of preSATP6-RF1-OMA1 interaction is indicated by an arrow. (b) Co-immunoprecipitation of total protein extracts from a crude mitochondria preparation isolated from transgenic suspension cells. Protein samples were separated by 12% SDS–PAGE before or after immunoprecipitation using anti-FLAG (input or IP, respectively). The separated proteins were blotted onto a membrane and probed with anti-FLAG ($\alpha$FLAG) or anti-preSATP6 ($\alpha$preSATP6).

It was possible that alteration occurred but was masked by other signal bands. Because alteration in the molecular mass of preSATP6-containing complexes is tightly associated with protein–protein binding between *preSatp6* and *RF1-Oma1*, we tested the protein–protein interaction between *preSatp6* and *atOma1* or *bvOma1* by co-immunoprecipitation. Total mitochondrial proteins from the transgenic suspension cells were precipitated with antibody against FLAG. The precipitates, however, did not react with preSATP6 antiserum (figure 4b), indicating that protein–protein binding between *preSatp6* and *atOma1* or *bvOma1* was below the limit of detection by this assay.

The *Oma1*-like genes form a small gene family in beets, unlike in the model plants Arabidopsis and rice [15,23]. Gene duplication is the most likely mechanism to explain the generation of this gene family. Two members of this family, *bvOma1* and *RF1-Oma1* (including *rf1-Oma1*), are ubiquitous among sugar beet lines. Whereas *bvOma1* occurs as a single copy gene in all lines examined so far, *RF1-Oma1* shows CNV among beet lines, as was reported previously [18,19,26]. In our previous study, an organizational comparison of beet *Rf1* alleles suggested that intra- and/or intergenic recombination played an important role in this variation [17]. We should point out that analogous variations occur in other

plant *Rf* loci encoding pentatrico peptide repeat proteins (PPRs) that participate in post-transcriptional regulation of genes responsible for CMS [12,34,35]. It is possible that beet *Rf1* and the PPR-type *Rf*s share evolutionary mechanisms.

The protein product of *RF1-Oma1* binds the preSATP6 protein and forms a 200 kDa protein complex, and suppression of CMS is tightly associated with this activity [16,17]. In this study, we found that such activity is absent from the protein products of *bvOma1* and *atOma1*, suggesting that *Oma1* was originally incapable of binding the preSATP6 protein. *RF1-Oma1* also differs from *bvOma1* and *atOma1* in its spatial expression pattern. We hypothesize that *RF1-Oma1* evolved through neofunctionalization of a duplicated *Oma1* gene. Furthermore, the differences in the spatial expression patterns of *bvOma1* and *atOma1* suggest the possibility that the evolution of *RF1-Oma1* might involve additional mechanisms, such as complementary degenerative mutations [36] in *RF1-Oma1* and other *Oma1*-like gene(s). However, these are only speculations, and further detailed studies are necessary to clarify the evolution of sugar beet *Rf1*.

Data accessibility. Our data are provided as electronic supplementary material.

Authors' contributions. T.A. and T.Ku. designed the study; T.A., H.S., T.Ka., Y.H., K.M. and K.K. carried out the molecular genetics work; H.M., Y.K. and T.Ku. developed plant materials; T.A., H.S., K.K. and T.Ku. analysed data; T.A. and T.Ku. drafted the manuscript. All authors gave final approval for publication.

Competing interests. There are no competing interests.

Funding. This work was supported in part by JSPS KAKENHI (grant no. 18K05564) (Y.H., H.M., K.K. and T.Ku.) and NARO Bio-oriented Technology Research Advancement Institution (BRAIN) (Research program on development of innovative technology) (grant no. 30001A) (H.M., Y.K., K.K. and T.Ku.). T.A. was a recipient of a JSPS Research Fellowship for Young Scientists (grant no. 16J01146).

Acknowledgements. Part of this study was done at Field Science Center for Northern Biosphere, Hokkaido University.

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
