## [Reviewer comments · Royal Society Open Science]

Review History

RSOS-190853.R0 (Original submission)

Review form: Reviewer 1

Is the manuscript scientifically sound in its present form?

Yes

Are the interpretations and conclusions justified by the results?

No

Is the language acceptable?

Yes

Do you have any ethical concerns with this paper?

No

Have you any concerns about statistical analyses in this paper?

No

Recommendation?

Major revision is needed (please make suggestions in comments)

Comments to the Author(s)

This manuscript reports the functional analysis of sugar beet homologues of the Oma1 yeast gene, one of which had been previously identified as an unusual Rf gene (Matsuhira et al, 2012), to enlighten molecular events that gave rise to the emergence of an Rf-functional copy of the Oma1 gene.

This study identifies several copies of Oma1 homologues in different sugar beet genomes and analyses the expression of the unique Arabidopsis Oma1 homologue (named AtOma1), its closest sugar beet homologue (named BvOma1) and the restorer copy (RF1-Oma1) in anthers during pollen development.

The main result convincingly indicates that AtOma1 and BvOma1 are not able to interact with the sterility protein, nor to modify the size of its complex, suggesting a neofunctionalization of the RF1-Oma1 copy, which is considered to be inactive for the primary protein quality control of OMA1 protease in yeast mitochondria.

These results lead the authors to propose that the RF1-Oma1 gene emerged after a duplication-degeneration-complementation of the sugar beet Oma1 gene for its expression pattern, followed by a neofunctionalization of RF1-Oma1 copies.

However, the evolutionary model proposed is largely speculative. The hypothesis concerning the duplication-degeneration-complementation evolution is based on disputable results (see below) of expression patterns, and only in male organs while the primary Oma1 function is expected to be important in any mitochondrially active cell type. Although the DDC model would indeed explain the copy number of this gene in beet genomes, conclusively addressing its role appears beyond the interest of this study, which lies in the unprecedentedly reported neofunctionalization of a "house-keeping" protease into a restorer gene for CMS.

Nevertheless, a closer insight into the phylogenetic relationships between sequences of Oma1 homologues would be necessary, as well as a conclusive result about the copy(ies) that actually fulfills the primary Oma1 protease function (by complementation of a yeast mutant for example). One remarkable result is the variability in copy number of Oma1 homologues amongst the genotypes. A comment on this finding would be relevant considering the similar characteristic previously found in Rf loci encoding PPRs (eg Kato et al, 2007).

An indication whether the expansion of Oma1 copy number is restricted to *Beta vulgaris* or extends in sister species would be of great interest.

As a general comment, I regret that the introduction does not give a better idea about the currently accepted evolutionary models for Rf genes. This would give an enlightening context to the reader.

For the sake of clarity, I suggest that all genotypes be described in a table, including TK-81, NK-198 and NK-305, as well as the two reference genomes KWS2320 and EL10, specifying their restoring status at the Rf1 locus, if known.

l. 250: give data source for this statement (Fig S3 for BvOma1, ? for RF1-Oma1).

l. 257. Sequence alignments and inferred phylogenetic tree should be used to designate an ancestral copy or homologue.

I have several concerns regarding the analysis of BvOma1 and RF1-Oma1 expression patterns.

- Oma1 copies are described in two sugar beet genomes, but expression pattern analysis were conducted in another one, characterized for the CMS/Rf study. The number of other Oma1 homologues is unclear in this genotype, although one homologue of Oma1 was not found.

- considering the presence of several homologues in sugar beet genomes, the specificity of primers for qRT-PCR should be thoroughly verified and shown.

- what is the relevance of measuring the amount of RNA of an inactive Rf gene (l.266-269)?

- from the examination of Fig 2, I do not pull the same conclusions as the authors for the expression of AtOma1 in tetrads (Fig 2A and l. 273), and microspores (Fig 2A and l.274-275): the

signal in microspores with the sense probe is more than "slight". I cannot see the ISH signal with RF1-Oma1 antisense probe in mother pollen cells (Fig 2C and l. 282). However, with the same probe, I can see a signal in the tapetum of the tetrad stage that is not mentioned in the text.
 - l. 304-306 the conclusion that AtOma1 and BvOma1 are not functionally equivalent based on their (possible) differences in expression pattern is very speculative. So is the necessity for an additional factor hypothesized l. 309-311.

Other comments:

A table for primers would facilitate the reading

indicate the position of primers on the sequence alignments

l. 192-202 elution of the IP from magnetic beads is not described.

The localization of the Oma1-carrying scaffolds on chromosome 3 should be given (in Table S1 for instance).

Fig3B legend l.483: if only transgene expressing cell lines were analyzed, what are lines NT?

l. 444 year is missing in the reference

Review form: Reviewer 2

Is the manuscript scientifically sound in its present form?

Yes

Are the interpretations and conclusions justified by the results?

No

Is the language acceptable?

Yes

Do you have any ethical concerns with this paper?

No

Have you any concerns about statistical analyses in this paper?

No

Recommendation?

Major revision is needed (please make suggestions in comments)

Comments to the Author(s)

The manuscript describes the characterization of a small gene family in sugar beet comprising and related to the Rf1 nuclear restorer of sugar beet CMS. There are a number of unusual features of CMS in sugar beet. Beet CMS is specified by a form of the atp6 gene that includes a presequence at the N-terminus that is cleaved from the mature ATP6 protein but accumulates in the mitochondrial inner membrane as part of a large complex. The size of this complex is reduced in plants containing a nuclear restorer gene (Rf1). Previous work from this research group has shown that Rf1 resembles the mitochondrial inner membrane protease Oma1 that is involved in mitochondrial quality control mechanisms. The Rf1 form, however, has sustained a mutation that likely eliminates its proteolytic function.

The analysis of the Oma1/RF1 gene family is relatively straightforward. It is of particular interest that a form of the gene encoding a protein without apparent proteolytic function has expanded through duplication events and serves, in one allelic form, as a novel restorer of male sterility.

This, of itself, is consistent with the authors' suggestion that the restorer evolved from a duplicated deficient form of *Oma1* that acquired a new function.

The experimental data of Fig. 3 are quite convincing. Too much emphasis is placed on the data of Fig. 2, however, particularly with respect to its implications regarding gene function and evolution. The functional equivalence, or lack thereof, of the same gene from two different species cannot be determined from the apparently different, but overlapping expression domains, as indicated in lines 304-306. This can only be answered by genetic complementation experiments (e.g. transforming an *Arabidopsis* knock-out with beet *Oma1*). Moreover, the differences in the staining of tetrad stage anthers between the sense and antisense probes of RF1-*Oma1* are marginal; it is not clear that the gene is expressed in, or only in, the tapetal layer.

Regarding the interpretation of the analysis of database sequences with *Oma1* homologues, a comparison of the upstream and downstream sequences of the various gene forms would be of interest and answer several questions that arise from this work. For example, do the different copies of rf1-*Oma1* in line EL10 have identical flanking sequences? What is the length of the duplicated segments surrounding the each gene copy?

The use of the designation RF1-*Oma* for both the restoring and non-restoring alleles of the restorer gene product previously identified by this group is extremely confusing. It is suggested that this nomenclature be revised and/or that some designation be incorporated to distinguish the two functionally distinguishable proteins. This could be accomplished most easily if the non-restoring allele product was designated as rf1-*Oma*.

The sequences of primers should be given in Table form, ideally as supplementary data.

The amino acid sequences of the different *Oma1* homologues are critical to an understanding of the manuscript and should be included as the first proper figure in the manuscript and not as a Supplementary figure.

Decision letter (RSOS-190853.R0)

21-Aug-2019

Dear Dr Kubo,

The editors assigned to your paper ("How did a duplicated gene copy evolve into a *restorer-of-fertility* gene in plant? A case of *Oma1*") have now received comments from reviewers.

Both reviewers are positive about publication of your paper. However, they both raise substantive issues with the current manuscript which will require careful attention. Please give particular attention to the points of the reviewers and the Associate Editor concerning areas where they highlight that you should reduce the claims of the paper. We would like you to revise your paper in accordance with the referee and Associate Editor suggestions which can be found below (not including confidential reports to the Editor). Please note this decision does not guarantee eventual acceptance.

Please submit a copy of your revised paper before 13-Sep-2019. Please note that the revision deadline will expire at 00.00am on this date. If we do not hear from you within this time then it will be assumed that the paper has been withdrawn. In exceptional circumstances, extensions

may be possible if agreed with the Editorial Office in advance. We do not allow multiple rounds of revision so we urge you to make every effort to fully address all of the comments at this stage. If deemed necessary by the Editors, your manuscript will be sent back to one or more of the original reviewers for assessment. If the original reviewers are not available, we may invite new reviewers.

- Data accessibility

<http://datadryad.org/submit?journalID=RSOS&manu=RSOS-190853>

- Competing interests

- Authors' contributions

- Acknowledgements

- Funding statement

Kind regards,

Alice Power

Editorial Coordinator

on behalf of Dr James Locke (Associate Editor) and Steve Brown (Subject Editor)

Associate Editor's comments (Dr James Locke):

Please revise the manuscript according to the suggestions of the two reviewers, paying special attention to the areas where they specify that you should reduce the claims of the paper.

Comments to Author:

Reviewers' Comments to Author:

Reviewer: 1

Comments to the Author(s)

This manuscript reports the functional analysis of sugar beet homologues of the Oma1 yeast gene, one of which had been previously identified as an unusual Rf gene (Matsuhira et al, 2012), to enlighten molecular events that gave rise to the emergence of an Rf-functional copy of the Oma1 gene.

This study identifies several copies of Oma1 homologues in different sugar beet genomes and analyses the expression of the unique Arabidopsis Oma1 homologue (named AtOma1), its closest sugar beet homologue (named BvOma1) and the restorer copy (RF1-Oma1) in anthers during pollen development.

The main result convincingly indicates that AtOma1 and BvOma1 are not able to interact with the sterility protein, nor to modify the size of its complex, suggesting a neofunctionalization of the

RF1-Oma1 copy, which is considered to be inactive for the primary protein quality control of OMA1 protease in yeast mitochondria.

These results lead the authors to propose that the RF1-Oma1 gene emerged after a duplication-degeneration-complementation of the sugar beet Oma1 gene for its expression pattern, followed by a neofunctionalization of RF1-Oma1 copies.

However, the evolutionary model proposed is largely speculative. The hypothesis concerning the duplication-degeneration-complementation evolution is based on disputable results (see below) of expression patterns, and only in male organs while the primary Oma1 function is expected to be important in any mitochondrially active cell type. Although the DDC model would indeed explain the copy number of this gene in beet genomes, conclusively addressing its role appears beyond the interest of this study, which lies in the unprecedentedly reported neofunctionalization of a "house-keeping" protease into a restorer gene for CMS.

Nevertheless, a closer insight into the phylogenetic relationships between sequences of Oma1 homologues would be necessary, as well as a conclusive result about the copy(ies) that actually fulfills the primary Oma1 protease function (by complementation of a yeast mutant for example). One remarkable result is the variability in copy number of Oma1 homologues amongst the genotypes. A comment on this finding would be relevant considering the similar characteristic previously found in Rf loci encoding PPRs (eg Kato et al, 2007).

An indication whether the expansion of Oma1 copy number is restricted to *Beta vulgaris* or extends in sister species would be of great interest.

As a general comment, I regret that the introduction does not give a better idea about the currently accepted evolutionary models for Rf genes. This would give an enlightening context to the reader.

For the sake of clarity, I suggest that all genotypes be described in a table, including TK-81, NK-198 and NK-305, as well as the two reference genomes KWS2320 and EL10, specifying their restoring status at the Rf1 locus, if known.

l. 250: give data source for this statement (Fig S3 for BvOma1, ? for RF1-Oma1).

l. 257. Sequence alignments and inferred phylogenetic tree should be used to designate an ancestral copy or homologue.

I have several concerns regarding the analysis of BvOma1 and RF1-Oma1 expression patterns.

- Oma1 copies are described in two sugar beet genomes, but expression pattern analysis were conducted in another one, characterized for the CMS/Rf study. The number of other Oma1 homologues is unclear in this genotype, although one homologue of Oma1 was not found.

- considering the presence of several homologues in sugar beet genomes, the specificity of primers for qRT-PCR should be thoroughly verified and shown.

- what is the relevance of measuring the amount of RNA of an inactive Rf gene (l.266-269)?

- from the examination of Fig 2, I do not pull the same conclusions as the authors for the expression of AtOma1 in tetrads (Fig 2A and l. 273), and microspores (Fig 2A and l.274-275): the signal in microspores with the sense probe is more than "slight". I cannot see the ISH signal with RF1-Oma1 antisense probe in mother pollen cells (Fig 2C and l. 282). However, with the same probe, I can see a signal in the tapetum of the tetrad stage that is not mentioned in the text.

- l. 304-306 the conclusion that AtOma1 and BvOma1 are not functionally equivalent based on their (possible) differences in expression pattern is very speculative. So is the necessity for an additional factor hypothesized l. 309-311.

Other comments:

A table for primers would facilitate the reading

indicate the position of primers on the sequence alignments

l. 192-202 elution of the IP from magnetic beads is not described.

The localization of the Oma1-carrying scaffolds on chromosome 3 should be given (in Table S1 for instance).

Fig3B legend l.483: if only transgene expressing cell lines were analyzed, what are lines NT?

l. 444 year is missing in the reference

Reviewer: 2

Comments to the Author(s)

The manuscript describes the characterization of a small gene family in sugar beet comprising and related to the Rf1 nuclear restorer of sugar beet CMS. There are a number of unusual features of CMS in sugar beet. Beet CMS is specified by a form of the *atp6* gene that includes a presequence at the N-terminus that is cleaved from the mature ATP6 protein but accumulates in the mitochondrial inner membrane as part of a large complex. The size of this complex is reduced in plants containing a nuclear restorer gene (Rf1). Previous work from this research group has shown that Rf1 resembles the mitochondrial inner membrane protease Oma1 that is involved in mitochondrial quality control mechanisms. The Rf1 form, however, has sustained a mutation that likely eliminates its proteolytic function.

The analysis of the Oma1/RF1 gene family is relatively straightforward. It is of particular interest that a form of the gene encoding a protein without apparent proteolytic function has expanded through duplication events and serves, in one allelic form, as a novel restorer of male sterility. This, of itself, is consistent with the authors' suggestion that the restorer evolved from a duplicated deficient form of Oma1 that acquired a new function.

The experimental data of Fig. 3 are quite convincing. Too much emphasis is placed on the data of Fig. 2, however, particularly with respect to its implications regarding gene function and evolution. The functional equivalence, or lack thereof, of the same gene from two different species cannot be determined from the apparently different, but overlapping expression domains, as indicated in lines 304-306. This can only be answered by genetic complementation experiments (e.g. transforming an Arabidopsis knock-out with beet Oma1. Moreover, the differences in the staining of tetrad stage anthers between the sense and antisense probes of RF1-Oma1 are marginal; it is not clear that the gene is expressed in, or only in, the tapetal layer.

Regarding the interpretation of the analysis of database sequences with Oma 1 homologues, a comparison of the upstream and downstream sequences of the various gene forms would be of interest and answer several questions that arise from this work. For example, do the different copies of rf1-Oma1 in line EL10 have identical flanking sequences? What is the length of the duplicated segments surrounding the each gene copy?

The use of the designation RF1-Oma for both the restoring and non-restoring alleles of the restorer gene product previously identified by this group is extremely confusing. It is suggested that this nomenclature be revised and/or that some designation be incorporated to distinguish the two functionally distinguishable proteins. This could be accomplished most easily if the non-restoring allele product was designated as rf1-Oma.

The sequences of primers should be given in Table form, ideally as supplementary data.

The amino acid sequences of the different Oma1 homologues are critical to an understanding of the manuscript and should be included as the first proper figure in the manuscript and not as a Supplementary figure.

Author's Response to Decision Letter for (RSOS-190853.R0)

See Appendix A.

Decision letter (RSOS-190853.R1)

08-Oct-2019

Dear Dr Kubo,

I am pleased to inform you that your manuscript entitled "How did a duplicated gene copy evolve into a *restorer-of-fertility* gene in a plant? The case of *Oma1*" is now accepted for publication in Royal Society Open Science.

Best regards,
Lianne Parkhouse
Royal Society Open Science
openscience@royalsociety.org

on behalf of Dr James Locke (Associate Editor) and Professor Steve Brown (Subject Editor)
openscience@royalsociety.org

Associate Editor Comments to Author (Dr James Locke):

The revision has addressed the reviewers' comments.

Follow Royal Society Publishing on Twitter: [@RSocPublishing](https://twitter.com/RSocPublishing)

Appendix A

Responses to Associate Editor

Please revise the manuscript according to the suggestions of the two reviewers, paying special attention to the areas where they specify that you should reduce the claims of the paper.

Thank you very much for your consideration. We are very glad to see that both reviewers consider our key results valuable, and we have made revisions according to their suggestions. We reduced the discussion of DDC but stressed neofunctionalization as a model for the evolution of *RF1-Oma1*, since that seems to be accepted by both reviewers. Our revisions are marked by yellow highlights in the revised manuscript, which was edited by a professional English editing service.

Responses to Reviewer 1:

*1-1 This manuscript reports the functional analysis of sugar beet homologues of the *Oma1* yeast gene, one of which had been previously identified as an unusual *Rf* gene (Matsuhira et al, 2012), to enlighten molecular events that gave rise to the emergence of an *Rf*-functional copy of the *Oma1* gene.*

*This study identifies several copies of *Oma1* homologues in different sugar beet genomes and analyses the expression of the unique *Arabidopsis* *Oma1* homologue (named *AtOma1*), its closest sugar beet homologue (named *BvOma1*) and the restorer copy (*RF1-Oma1*) in anthers during pollen development. The main result convincingly indicates that *AtOma1* and *BvOma1* are not able to interact with the sterility protein, nor to modify the size of its complex, suggesting a neofunctionalization of the *RF1-Oma1* copy, which is considered to be inactive for the primary protein quality control of *OMA1* protease in yeast mitochondria.*

*These results lead the authors to propose that the *RF1-Oma1* gene emerged*

after a duplication-degeneration-complementation of the sugar beet Oma1 gene for its expression pattern, followed by a neofunctionalization of RF1-Oma1 copies.

However, the evolutionary model proposed is largely speculative. The hypothesis concerning the duplication-degeneration-complementation evolution is based on disputable results (see below) of expression patterns, and only in male organs while the primary Oma1 function is expected to be important in any mitochondrially active cell type. Although the DDC model would indeed explain the copy number of this gene in beet genomes, conclusively addressing its role appears beyond the interest of this study, which lies in the unprecedentedly reported neofunctionalization of a "house-keeping" protease into a restorer gene for CMS.

Thank you very much for your careful reading of our manuscript. As you suggested, we have reduced our discussion of the DDC model in the revised manuscript (lines 294-315).

1-2 Nevertheless, a closer insight into the phylogenetic relationships between sequences of Oma1 homologues would be necessary, as well as a conclusive result about the copy(ies) that actually fulfills the primary Oma1 protease function (by complementation of a yeast mutant for example).

We are interested in differences between the molecular evolution of *RF1-Oma1* and *bvOma1* (see our responses below to points 1-3, 1-4, and 1-8). This is an ongoing project that includes analyses of nucleotide substitution patterns and detailed phylogenetic analyses of the plants. We agree that our manuscript lacks a functional analysis of *bvOma1*, although *bvOma1* is a likely orthologue of *atOma1*, because the other *Oma1*-like genes appear to have critical mutations (e. g., the Zn-binding motifs do not follow consensus). We modified our description to clarify this (lines 214-216).

1-3 One remarkable result is the variability in copy number of Oma1 homologues amongst the genotypes. A comment on this finding would be relevant considering the similar characteristic previously found in Rf loci encoding PPRs (eg Kato et al, 2007).

This has also attracted our attention because it implies that a common evolutionary mechanism may be involved in the PPR and non-PPR *Rf* genes (cf. Matsuhira et al. Genetics 2012). We added a sentence to point out this organizational similarity (lines 301-304).

1-4 An indication whether the expansion of Oma1 copy number is restricted to Beta vulgaris or extends in sister species would be of great interest.

We also think this is a very important point, and we are now investigating the *Oma1*-like genes in *Beta* plants. The results will appear elsewhere.

1-5 As a general comment, I regret that the introduction does not give a better idea about the currently accepted evolutionary models for Rf genes. This would give an enlightening context to the reader.

We added to the Introduction a brief description of the currently accepted evolutionary models for CMS and the *Rf* genes (lines 53-58).

1-6 For the sake of clarity, I suggest that all genotypes be described in a table, including TK-81, NK-198 and NK-305, as well as the two reference genomes KWS2320 and EL10, specifying their restoring status at the Rf1 locus, if known.

We summarized our information about the plant materials in Table 1.

1-7 1. 250: give data source for this statement (Fig S3 for BvOma1, ? for RF1-Oma1).

We added a sentence to provide the source (line 237-239).

1-8 1. 257. Sequence alignments and inferred phylogenetic tree should be used to designate an ancestral copy or homologue.

The *atOma1* gene is likely an *Oma1* orthologue because it is the only copy of *Oma1* in the Arabidopsis genome. The *bvOma1* gene is the best candidate for the sugar beet orthologue because it is preserved among beet lines and its Zn-binding motif in the peptidase M48 domain follows the consensus sequence. No other *Oma1*-like copy in the sugar beet genome has the consensus Zn-binding motif. But we avoided using the term “ancestral gene” in the revised manuscript (line 245), and we will provide a phylogenetic tree of plant *Oma1* sequences elsewhere (see also our responses to points 1-2 to 1-4).

1-9 I have several concerns regarding the analysis of BvOma1 and RF1-Oma1 expression patterns.

- Oma1 copies are described in two sugar beet genomes, but expression pattern analysis were conducted in another one, characterized for the CMS/Rf study. The number of other Oma1 homologues is unclear in this genotype, although one homologue of Oma1 was not found.

- considering the presence of several homologues in sugar beet genomes, the specificity of primers for qRT-PCR should be thoroughly verified and shown.

Our plant material for RT-PCR was TA-33BB-0, which lacks LOC104906603. This was clarified in Fig. S3C (and lines 251-252). We verified that our primer sets discriminate between *RF1-Oma1* and *bvOma1* (Fig. S1).

1-10 - what is the relevance of measuring the amount of RNA of an inactive Rf gene (l. 266-269)?

The *rf1-Oma1* gene in TA-33BB-0 is identical to that in KWS2320, and we added this information to the revised text (lines 255-256). Although the recessive *rf* alleles are unable to restore fertility, their ORFs remain intact and transcribed, suggesting that they are functional at the molecular level (Kitazaki et al. Plant J 2015; this study). Under the DDC model, which we hypothesize to be involved in *Rf* evolution, a recessive *rf* gene would remain functional and complement another duplicated gene, for the accomplishment of their ancestral function.

1-11 - from the examination of Fig 2, I do not pull the same conclusions as the authors for the expression of AtOma1 in tetrads (Fig 2A and l. 273), and microspores (Fig 2A and l. 274-275): the signal in microspores with the sense probe is more than "slight". I cannot see the ISH signal with RF1-Oma1 antisense probe in mother pollen cells (Fig 2C and l. 282). However, with the same probe, I can see a signal in the tapetum of the tetrad stage that is not mentioned in the text.

We are sorry for the confusion. The relevant descriptions have been modified (lines 264-265 and 273-274). High quality images are provided in a new version of the figure (Fig. 3 in the revised manuscript) to show the signal. We have clarified that the ISH signal of *RF1-Oma1* was detected in the tetrad stage of the tapetum (lines 271-272).

1-12 - 1. 304-306 the conclusion that AtOma1 and BvOma1 are not functionally equivalent based on their (possible) differences in expression pattern is very speculative. So is the necessity for an additional factor hypothesized 1. 309-311.

We have reduced the speculative discussion in the revised manuscript (lines 294-315).

1-13 A table for primers would facilitate the reading indicate the position of primers on the sequence alignments

We added a new table to summarize the primers used in the study (Table S1). The positions of primers are shown in Fig. S2.

1-14 1. 192-202 elution of the IP from magnetic beads is not described.

The IP was eluted by boiling in the SDS sample buffer. This information was added (lines 186-187).

1-15 The localization of the Oma1-carrying scaffolds on chromosome 3 should be given (in Table S1 for instance).

The contigs of KWS2320 were assigned to chromosome 3 because they include DNA markers previously assigned to chromosome 3 by genetic linkage analysis. However, the physical distance between the contigs is currently unknown because of a gap. We can only infer their relative positions from the EL10 data, which is the highest quality at present. This was clarified in the revised manuscript (lines 225-226). We added the assigned chromosome number

of the contigs in Table S1.

1-16 Fig3B legend l. 483: if only transgene expressing cell lines were analyzed, what are lines NT?

We deleted this sentence.

1-17 l. 444 year is missing in the reference

We added the year of publication (line 443).

Responses to Reviewer 2

2-1 The manuscript describes the characterization of a small gene family in sugar beet comprising and related to the Rf1 nuclear restorer of sugar beet CMS. There are a number of unusual features of CMS in sugar beet. Beet CMS is specified by a form of the atp6 gene that includes a presequence at the N-terminus that is cleaved from the mature ATP6 protein but accumulates in the mitochondrial inner membrane as part of a large complex. The size of this complex is reduced in plants containing a nuclear restorer gene (Rf1). Previous work from this research group has shown that Rf1 resembles the mitochondrial inner membrane protease Omal that is involved in mitochondrial quality control mechanisms. The Rf1 form, however, has sustained a mutation that likely eliminates its proteolytic function. The analysis of the Omal/RF1 gene family is relatively straightforward. It is of particular interest that a form of the gene encoding a protein without apparent proteolytic function has expanded through duplication events and serves, in one allelic form, as a novel restorer of male sterility. This, of itself, is consistent with the authors' suggestion that the

restorer evolved from a duplicated deficient form of Oma1 that acquired a new function.

The experimental data of Fig. 3 are quite convincing.

Thank you very much for your comments.

2-2 Too much emphasis is placed on the data of Fig. 2, however, particularly with respect to its implications regarding gene function and evolution. The functional equivalence, or lack thereof, of the same gene from two different species cannot be determined from the apparently different, but overlapping expression domains, as indicated in lines 304-306. This can only be answered by genetic complementation experiments (e.g. transforming an Arabidopsis knock-out with beet Oma1.

We reduced our discussion of gene function inferred from the data of temporal and spatial expression pattern (lines 294-315).

2-3 Moreover, the differences in the staining of tetrad stage anthers between the sense and antisense probes of RF1-Oma1 are marginal; it is not clear that the gene is expressed in, or only in, the tapetal layer.

We have provided higher quality images in a new version of the figure (Fig. 3 in the revised manuscript). The expression patterns were described in more detail in the revised manuscript (lines 271-274).

2-4 Regarding the interpretation of the analysis of database sequences with Oma 1 homologues, a comparison of the upstream and downstream sequences of the various gene forms would be of interest and answer several questions that arise from this work. For example, do the different copies of rf1-Oma1

in line EL10 have identical flanking sequences? What is the length of the duplicated segments surrounding the each gene copy?

We are also very interested in the organizational variation within the *Rf1* locus. In previous papers we compared sequences within the locus (both coding and flanking), and the results suggest that intergenic recombination played a major role in the variation. We added this information to the revised manuscript (lines 299-301).

2-5 The use of the designation RF1-Oma for both the restoring and non-restoring alleles of the restorer gene product previously identified by this group is extremely confusing. It is suggested that this nomenclature be revised and/or that some designation be incorporated to distinguish the two functionally distinguishable proteins. This could be accomplished most easily if the non-restoring allele product was designated as rf1-Oma.

We are sorry for your confusion. We have modified the paper according to your suggestion (lines 222-223 and throughout the manuscript).

2-6 The sequences of primers should be given in Table form, ideally as supplementary data.

We listed the primer sequences in Table S1.

2-7 The amino acid sequences of the different Oma1 homologues are critical to an understanding of the manuscript and should be included as the first proper figure in the manuscript and not as a Supplementary figure.

We have added a new Fig. 2, based on the former Fig. S1.